# Experimental Research of Triple Inertial Navigation System Shearer Positioning

**DOI:** 10.3390/mi14071474

**Published:** 2023-07-23

**Authors:** Cheng Lu, Shibo Wang, Kyoosik Shin, Wenbin Dong, Wenqi Li

**Affiliations:** 1School of Mechanical Engineering, Anhui Science and Technology University, Chuzhou 233100, China; luch@ahstu.edu.cn; 2School of Mechanical and Electrical Engineering, China University of Mining & Technology, Xuzhou 221116, China; 3Department of Mechatronics Engineering, Hanyang University, Ansan 15588, Gyeonggi-do, Republic of Korea; kss_1212@163.com (K.S.); wuhui0514@163.com (W.D.); liwenqi@hanyang.ac.kr (W.L.)

**Keywords:** shearer, TINS, multi-INS, positioning method, coal mine-intelligent

## Abstract

In order to improve the positioning accuracy of shearers, the overground experimental device based on the positioning model of TINS (Triple Inertial Navigation System) was built. The influence of TINS installation parameters on positioning accuracy was discussed through two sets of experiments: the inter-INS (Inertial Navigation System) distances influence experiments and the tri-INS plane spatial position influence experiments. The results show that the positioning accuracy of the shearer is improved to a different extent under the two sets of experimental conditions. When the inter-INS distances are 0.2 m, the positioning accuracy is the highest and the positioning accuracy improvement effect is also the best. When the negative plane α_3_ is 45°, the positioning accuracy is the highest, and the positioning accuracy improvement effect is also the best. The analysis shows that the main factor affecting the positioning accuracy is the precision of the evaluated values outputs of TINS from EKF (Extended Kalman Filter). Considering the positioning accuracy, equipment installation convenience and so on, the optimum installation parameters are 90° (horizontal installation) α_3_ for the positive plane and 0.2 m inter-INS distances.

## 1. Introduction

Intelligent and unmanned mining offers the prospect of safe and efficient mining [1]. The key to solving the core problem of the current development of intelligent mining is the real-time and accurate positioning of the shearer in the seam space of coal [2].

The INS positioning of shearers is an independent positioning technology that has been widely recognized by scientific research institutions and coal mining equipment manufacturers [3]. Although it has many technical advantages, it lacks the ability to eliminate and reduce the accumulated errors of inertial sensors over time. As a result, the method of improving INS positioning accuracy has become a research hotspot all over the world. As early as 2001, Australia’s CSIRO (Commonwealth Scientific and Industrial Research Organization) began to study the integrated inertial navigation system of shearers, which was a Doppler velocity measurement radar with Zero-Speed Correction Technology [4,5]. In the later study, the scholar Ralston [6] from the aforementioned organization designed a kind of Kalman filter algorithm to optimize the positioning accuracy of a shearer based on the closed paths formed by the shearer’s moving trajectory. Chinese scholars have also undertaken a variety of research on improving INS positioning accuracy of the shearer. From 2016 to now, the scientific research team of the authors has proposed Shearer Positioning Technology Combined with INS and Coder [7], Shearer INS Positioning Dynamic Zero-Speed Correction Technology [8], Dynamic Precise Positioning Technology Based on Closed Path Optimal Estimation Model [9], Dual Inertial Navigation Positioning Technology of Shearer [10], Multiple Inertial Navigation Redundancy Positioning Technology of Shearer [11], Shearer Positioning Technology Based on Heterogeneous Multi-source Information Fusion [12], Inertial Navigation and Ultra-broadband Fusion Positioning Technology of Shearer [13,14], Accuracy Improvement Method Based on Shearer Kinematic Constraint [15], Two-Point Method Based Deviation Compensation Algorithm Based on Two Points [2], and so on. The measurement error and installation deviation of INS have been investigated systematically.

Although previous scholars have conducted research on various methods to improve the INS positioning accuracy of shearers, they have mainly focused on aspects such as combining INS with other sensors to form an integrated navigation system or exploiting the kinematic constraints. The analysis of the reasons affecting the INS positioning accuracy mainly focused on the installation errors or inertial sensor errors [6,16,17]. Few scholars have conducted their research on multi-INS positioning technology of shearers and analysis of reasons influencing multi-INS positioning accuracy. However, the research of multi-INS are widespread in the aviation industry and other fields. Liu Zhi [18] took the differences of attitude angles of two INS as the observed values and took the zero velocity errors of accelerometers as the estimated values to realize the dynamic correction of accelerometer errors. Zhang Linlin [19] corrected the velocity error to some extent by using two or more INS damping networks of ships. To improve the INS accuracy of civil aircraft, Bai Junqiang [20] proposed an optimal navigation solution with three parallel INS. Through theoretical analysis and simulations, they concluded that the positioning accuracy could be improved under both normal operating conditions and fault conditions. Si [21] integrated the main INS positioning information and the INS positioning information of airborne weapons of fighter aircraft, thus improving the positioning accuracy of the weapon.

It can be seen from the above references that multi-INS technology is a feasible way of improving positioning accuracy. Compared with satellite, infrared, laser, ultrasonic and other positioning methods, multi-INS technology can improve the positioning accuracy by INS itself, which is more suitable for the complex coal mine environment with no satellite signal and high dust. In this paper, the TINS positioning model of shearers was investigated and the influences of installation parameters on the positioning accuracy were analyzed through experiments. In addition, the optimum installation parameters of TINS were proposed considering various factors. The experimental conclusions can provide some guidance for the application of multi-INS technology in the field of shearer positioning.

## 2. The TINS Positioning Principle of Shearers

The installation of the three INS on the shearer is shown in Figure 1. The geometric center of the upper surface of the shearer is defined as the origin (O_b_) of the shearer coordinate system (b system). INS-1 is installed at the origin of the shearer coordinate system, so the position of INS-1 represents the position of shearer. α is the angle between the Z_b_ axis and the line defined by INS-1 and INS-3. The geographic coordinate system (ENU, East-North-Up) is selected as the navigation coordinate system (n-system). The coder is installed on the axle of the shearer’s walking mechanism.

The composition and working mechanism of the TINS is shown in Figure 2. The three INS collect attitude angle values of shearer at the same time, then the transform cosine matrix from b-system to n-system is calculated (as shown in Equation (1)). The coder collects the displacements of the shear in b-system. It is converted to n-system by the transform cosine matrix (as shown in Equation (2)) and accumulated to the initial position of each INS in n-system (as shown in Equation (3)). Thus, the position values of the three INS in n-system are obtained. Those positioning values are used as state values (as shown in Equation (4)) and the inter-INS distances are used as observed values (as shown in Equation (5)). More accurate positioning value outputs are obtained from the EKF. The state equation and observed equation are shown in Equation (6) and Equation (8), respectively.
(1)$$C_{b}^{n}=\left[\begin{array}{ccc} \;\cos\gamma_{i}\cos\varphi_{i}+\sin\gamma_{i}\sin\theta_{i}\sin\varphi_{i} & \cos\theta_{i}\sin\varphi_{i} & \sin\gamma_{i}\cos\varphi_{i}-\cos\gamma_{i}\sin\theta_{i}\sin\varphi_{i}\\ -\cos\gamma_{i}\sin\varphi_{i}+\sin\gamma_{i}\sin\theta_{i}\cos\varphi_{i} & \cos\theta_{i}\cos\varphi_{i} & -\sin\gamma_{i}\sin\varphi_{i}-\cos\gamma_{i}\sin\theta_{i}\cos\varphi_{i}\\ -\sin\gamma_{i}\cos\theta_{i} & \sin\theta_{i} & \cos\gamma_{i}\cos\theta_{i} \end{array}\right]$$

*φ*, *θ* and *γ* are the heading, pitch and rolling measured by INS-*i*.
(2)$$\Delta S_{i}^{n}=\left[\begin{array}{c} \cos\theta_{i}\sin\varphi_{i}\\ \cos\theta_{i}\cos\varphi_{i}\\ \sin\theta_{i} \end{array}\right]s$$

*φ* and *θ* are the heading and pitch collected by INS-*i*, ${∆S}_{i}^{n}$ is the displacement of INS-*i* in n-system and *s* is the shearer’s displacement collected by the coder.
(3)$$P_{i}^{n}(k)={\left[X_{i},Y_{i},Z_{i}\right]}_{k}^{T}=P_{i}^{n}(0)+\sum_{j=1}^{k}\Delta S_{i}^{n}(j)$$

$P_{i}^{n}$(*k*) is the position of INS-*i* in n-system at *k* time. $P_{i}^{n}$(0) is the position of INS-*i* in n-system at 0 time. *X_i_*, *Y_i_* and *Z_i_* is the coordinate value of north, east and up of INS-*i* in n-system.
(4)$$X={\left(\begin{array}{ccccccccc} x_{1} & y_{1} & z_{1} & x_{2} & y_{2} & z_{2} & x_{3} & y_{3} & z_{3} \end{array}\right)}^{T}$$

*x_i_*, *y_i_* and *z_i_* (*i* = 1, 2, 3) are the positioning in n-system of INS-*i*.
(5)$$Z={\left[r_{12},r_{13},r_{23}\right]}^{T}$$

*r_ij_* is the inter-INS distance between INS-*i* and INS-*j*.
(6)$$X\left(k+1\right)=X(k)+B(k)v(k)T+W(k)$$

*B*(*k*) is given by Equation (7), where *v*(*k*) is the shearer’s instantaneous velocity computed by the coder, *T* is the sampling period of INS and *W*(*k*) is the state noise at time *k*.
(7)$$B(k)=\left[\begin{array}{c} \cos\theta_{1}(k)\sin\varphi_{1}(k)\\ \cos\theta_{1}(k)\cos\varphi_{1}(k)\\ \sin\theta_{1}(k)\\ \cos\theta_{2}(k)\sin\varphi_{2}(k)\\ \cos\theta_{2}(k)\cos\varphi_{2}(k)\\ \sin\theta_{2}(k)\\ \cos\theta_{3}(k)\sin\varphi_{3}(k)\\ \cos\theta_{3}(k)\cos\varphi_{3}(k)\\ \sin\theta_{3}(k) \end{array}\right]$$

*r_ij_* is the inter-INS distance between INS-*i* and INS-*j*,
(8)Z(k)=H⋅X(k)+V(k)

*V*(*k*) is the observed noise at time *k*; *H* is the Jacobian matrix (shown in Equation (9)),
(9)$$H=\left[\begin{array}{ccccccccc} \frac{x_{1}-x_{2}}{r_{12}} & \frac{y_{1}-y_{2}}{r_{12}} & \frac{z_{1}-z_{2}}{r_{12}} & \frac{x_{2}-x_{1}}{r_{12}} & \frac{y_{2}-y_{1}}{r_{12}} & \frac{z_{2}-z_{1}}{r_{12}} & 0 & 0 & 0\\ \frac{x_{1}-x_{3}}{r_{13}} & \frac{y_{1}-y_{3}}{r_{13}} & \frac{z_{1}-z_{3}}{r_{13}} & 0 & 0 & 0 & \frac{x_{3}-x_{1}}{r_{13}} & \frac{y_{3}-y_{1}}{r_{13}} & \frac{z_{3}-z_{1}}{r_{13}}\\ 0 & 0 & 0 & \frac{x_{2}-x_{3}}{r_{23}} & \frac{y_{2}-y_{3}}{r_{23}} & \frac{z_{2}-z_{3}}{r_{23}} & \frac{x_{3}-x_{2}}{r_{23}} & \frac{y_{3}-y_{2}}{r_{23}} & \frac{z_{3}-z_{2}}{r_{23}} \end{array}\right]$$

## 3. Experimental Device and Methods

### 3.1. Experimental Device

The experimental device is composed of three INS with the same type and a manual turntable, coder, 12 V DC power source, serial ports hub and upper computer. The three INS were manufactured by Beijing CNSENS Technology Co., Ltd. (Beijing, China) and typed IMU680-G. The main performance indexes of their gyroscopes are shown in Table 1. All three INS are connected to serial ports hubs with an RS422 interface. The coder is connected to serial ports hub with an RS485 interface. The two interfaces are converted to a USB interface and connected to the upper computer via a serial ports hub. All the hardware is installed on the mobile vehicle by means of a shelf with three levels (as shown in Figure 3). The relative position of the three INS can be changed by adjusting the height of the second level. The GPS-RTK positioning system is also installed on the third level. The track of the high-precision GPS-RTK mobile station is taken as the real track of the experimental device. It can be seen from the TINS positioning model of the shearer that INS-1 is located at the geometric center of the shearer’s upper surface, so it can represent the shearer’s positioning value. Therefore, the GPS-RTK mobile station is installed close to INS-1 as a reference of the real position of the experimental device.

### 3.2. Methods

A heading angle of 40° north to east was chosen for ease of operation. The experimental device was moved in a straight line along the gaps between the floor tiles (as shown in Figure 4) to simulate the movement of the shearer along the coal wall. The initial conditions of the experiments were set as follows: the moving speed of the experimental device was set to 1.0 m/s–1.2 m/s, the moving distance of the experimental device was about 60 m and the three INS and the GPS-RTK were started at the same time to synchronize the running time of the two positioning systems. Through the following two sets of experiments, the influence of the inter-INS distances and the spatial position of the tri-INS plane on the positioning accuracy of the shearer was discussed. The experimental procedure is shown in Figure 5. The installation parameters of the experimental device are given in Table 2.

In order to investigate the influence of the inter-INS distances, the three INS were arranged horizontally, and the inter-INS distances were set as *r*_12_ = *r*_13_ = *h*, as shown in Figure 6a. The *h* values chosen were 0.1 m, 0.2 m, 0.28 m and 0.38 m, respectively.

To investigate the influence of the Tri-INS plane spatial position, the three INS were arranged at appropriate positions on the second and third layers of the shelf. The height of the second layer was adjusted so that *α*_3_ varied in the range of 0~360° (as shown in Figure 6b), so that the planes determined by the three INS covered the entire three-dimensional space. If the *X_b_* coordinate of INS-3 is positive, it is defined as the positive plane. Otherwise, it is defined as the negative plane.

The Spherical Probability Error (SEP), which is commonly used in the field of navigation and positioning, was used to evaluate the positioning accuracy. The SEP is calculated according to Equation (10).
(10)SEP=0.51(δx+δy+δz)

*σx*, *σy* and *σz* are the root mean square of the positioning errors in the east, north and upward directions, respectively.

## 4. Results and Discussions

### 4.1. Inter-INS Distances Influence Experiments

Figure 7 shows the TINS positioning trajectories and the single-INS positioning trajectories in the north–east plane and north–up plane of Experiment No.4 in Table 2. As can be seen from the figures, the TINS positioning trajectories are closer to the real trajectories than the single-INS positioning trajectories in the above two planes. The single-INS positioning trajectories fluctuate irregularly in the two planes, and the TINS positioning trajectories curves are relatively smoother.

Figure 8 shows the real-time positioning errors of the inter-INS distances influence experiments in Table 2. The maximum errors of single-INS in east, north and up are 10.29 m, 11.12 m and 10.53 m, respectively, while the maximum errors of TINS are 3.28 m, 3.79 m and 0.44 m, respectively. The maximum error of TINS is obviously lower than that of single-INS. The SEP of single-INS positioning is 7.77 m, and that of TINS positioning is 2.38 m. The accuracy of the TINS is 69.35% better than that of single-INS.

In Experiments No.1, No.2 and No.3 in Table 2, the SEP of single-INS positioning is 6.94 m, 6.67 m and 7.67 m, respectively, while the SEP of TINS positioning is 3.37 m, 1.58 m and 3.38 m (as shown in Figure 9). Compared to the single-INS positioning, the TINS positioning accuracy is improved by 53.77%, 76.31% and 55.93%, respectively. The positioning accuracy of the shearer is improved when the inter-INS distances take different values. The positioning accuracy is the highest and the positioning accuracy improvement effect is also the best when the inter-INS distances are 0.2 m.

### 4.2. The Tri-INS Plane Spatial Position Influence Experiments

In the tri-INS plane spatial position influence experiments, the SEP of single-INS positioning in the positive plane is 7.77 m, 6.57 m, 6.76 m, 7.64 m and 6.67 m, respectively, while the SEP of TINS positioning is 2.38 m, 2.52 m, 2.24 m, 3.73 m and 2.91 m, respectively (as shown in Figure 10a). The positioning accuracy of the TINS is improved by 69.36%, 61.64%, 66.86%, 51.17% and 56.37% compared to the single-INS. In the positive plane, the positioning accuracy of TINS is highest when *α*_3_ is 90°, and the positioning accuracy improvement effect is best when α_3_ is 0°. In the negative plane, the SEP of single-INS positioning is 6.19 m, 6.76 m, 6.16 m, 6.48 m and 5.93 m, respectively, while the SEP of TINS positioning is 2.49 m, 1.72 m, 3.00 m, 2.20 m and 2.22 m, respectively (as shown in Figure 10b). The positioning accuracy of TINS is improved by 59.77%, 74.56%, 51.30%, 66.05% and 62.56% compared to the single-INS. The positioning accuracy is the highest and the positioning accuracy improvement effect is also the best when α_3_ is 45°.

The experimental results shown in Figure 9 and Figure 10 as well as the analysis results of the previous simulations carried out by the authors of [10] show that the positioning accuracy of the TINS is higher than that of the single-INS under various working conditions. The real triangle formed by the three INS can be approximated as a rigid body. The lengths of the three sides of the triangle are sent to EKF as the observed quantities, which plays a role in limiting the accumulation of INS errors. When the inter-INS distances are 0.2 m and *α*_3_ is 45° in the negative plane, the positioning accuracy is the highest and the positioning accuracy improvement effect is also the best. However, considering the positioning accuracy and the convenience of the equipment installation, the positive plane with α_3_ 90° (horizontal installation) is optimum.

The estimated outputs of the three INS positionings were sampled every 20 s, using the EKF in Experiment No.7 of Table 2, and were plotted as a triangle (estimated triangle) in three-dimensional space at each sampling. It was projected on a north–east plane and east–up plane as shown in Figure 11. Time 0 was the initial time at which the projections in the two planes completely coincided with the real position projections of the three INS. In the later times, the shape of projections in the two planes changed to some extent with the movement of the experimental device. Figure 12 is the real-time variation diagram of the estimated triangle side length under this experimental condition. It can be seen from the figure that the estimated triangle side lengths have changed to some extent, but it is not as obvious as the change in projections in the two planes. It indicates that the shape of the triangle in three-dimensional space has not changed that much. The main reason for the change in the shape of the projections in the two planes is that the normal vector of the tri-INS plane deviated to a large extent. The secondary reason is the change in shape of the estimated triangle. It can be seen from the EKF model of the TINS positioning of the shearer that the velocity of the shearer is collected by the coder, so the main factors affecting the positioning accuracy are the accuracy of the three INS positioning estimated outputs and the accuracy of the observed values at the last moment. The deviation of the plane normal vector of the estimated triangle and its shape change mainly depend on the output of the estimated values of the three INS at the last moment from the EKF. When the three INS are installed, the real inter-INS distances will not change, since the accuracy of the observed values has been determined. Therefore, the main factor influencing the positioning accuracy is the accuracy of the estimated values of the three INS positions at the last moment.

## 5. Conclusions

In this paper, an experimental device was built according to the mathematical model of TINS positioning of shearers, and the influence of inter-INS distances and the spatial position of the tri-INS plane was analyzed experimentally. The results show that the positioning accuracy of TINS is higher than that of single-INS and the positioning trajectories are smoother under various experimental conditions. The main reason affecting the positioning accuracy is the output accuracy of the three INS position estimated values from the EKF at the last moment. Considering positioning accuracy of the shearer and the convenience of the equipment installation, the installation parameters of a positive plane with α_3_ 90° (horizontal installation) and 0.2 m inter-INS distances are the most suitable. Although the experimental device could not be operated for a long time due to the limitations of the overground experimental conditions, the experiments in this paper and the author’s previous simulations [10] show that the positioning accuracy of multi-INS is higher than that of single-INS with the same type. In addition, the purpose of this paper is to verify the effectiveness of the proposed method for improving the positioning accuracy of shearers. If the accuracy improvement effect is verified, high-precision INS will be used in real shearer experiments to meet the application requirements. It is also necessary to consider the influence of the vibration of running the shearer on the positioning accuracy during underground experiments in mines. Whether vibration isolation technology should be used to eliminate or reduce these influences needs further study. Multi-INS positioning technology not only plays a positive role in improving positioning accuracy, but the system also has the advantages of strong fault tolerance and good reliability. This research on multi-INS positioning technology of shearer has a good prospect of application in the process of intelligent coal mining.

## Figures and Tables

**Figure 1 micromachines-14-01474-f001:**
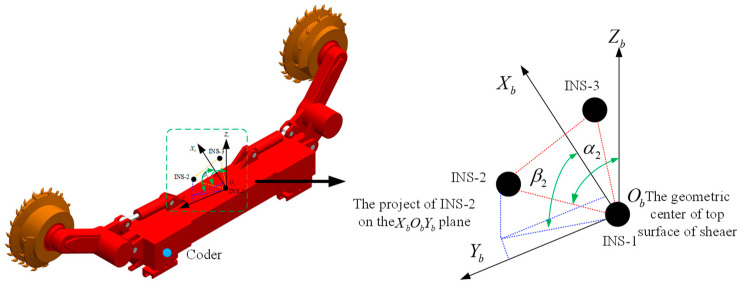
Installation of TINS.

**Figure 2 micromachines-14-01474-f002:**
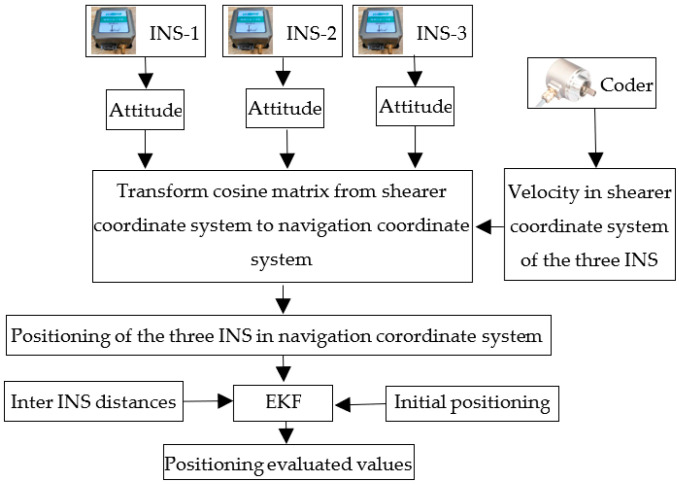
The composition and working mechanism of TINS.

**Figure 3 micromachines-14-01474-f003:**
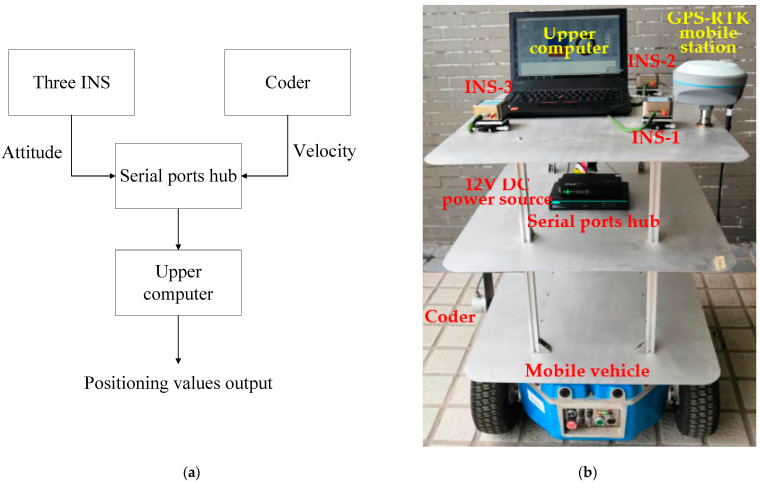
Experimental device. (**a**) Schematic diagram; (**b**) Hardware.

**Figure 4 micromachines-14-01474-f004:**
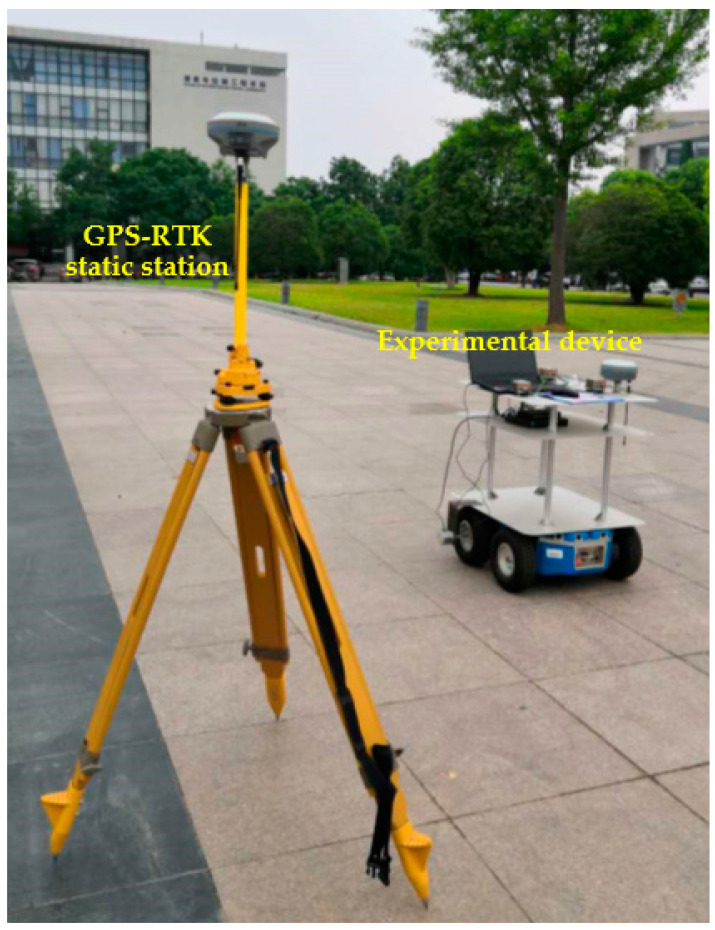
Schematic diagram of the experimental methods.

**Figure 5 micromachines-14-01474-f005:**
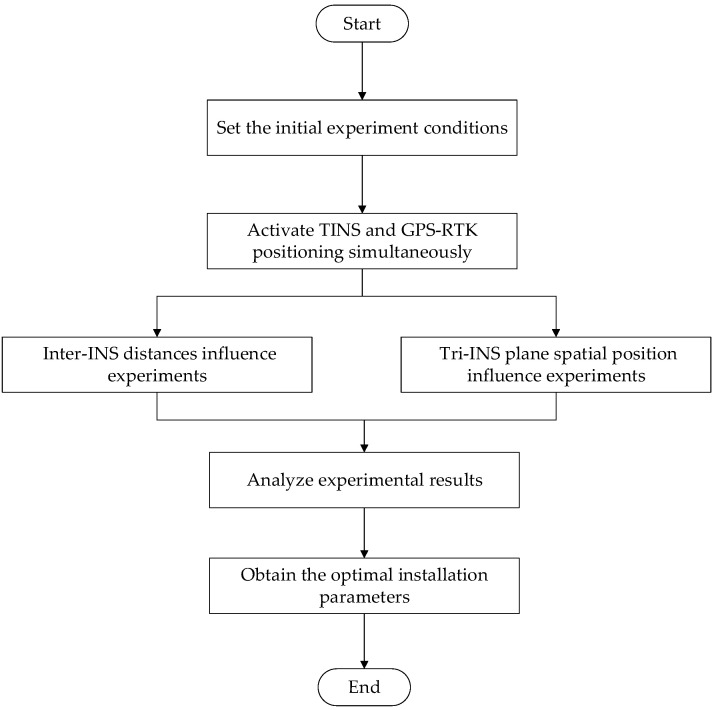
The experimental flow chart.

**Figure 6 micromachines-14-01474-f006:**
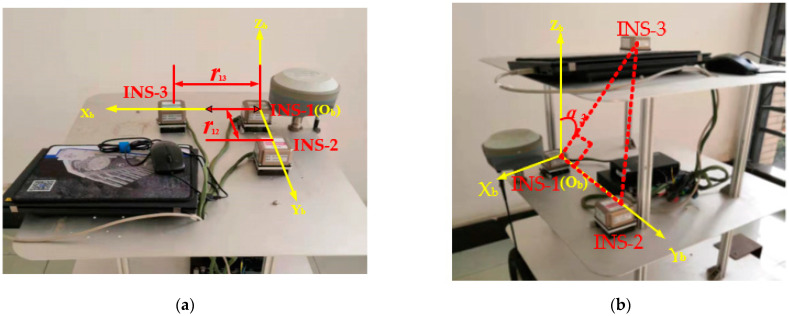
Schematic diagram of TINS. (**a**) Inter-INS distances influence experiments; (**b**) Tri-INS plane spatial position influence experiments.

**Figure 7 micromachines-14-01474-f007:**
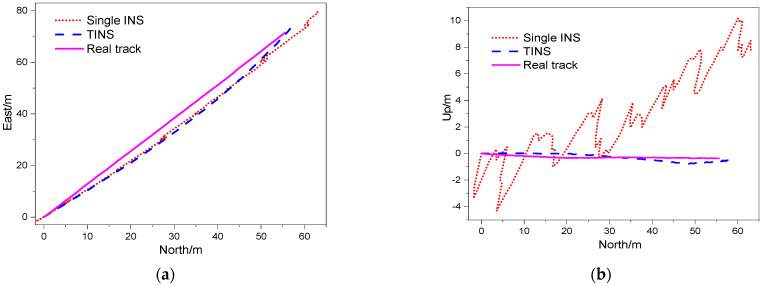
TINS positioning trajectories. (**a**) East-north plane. (**b**) North-up plane.

**Figure 8 micromachines-14-01474-f008:**
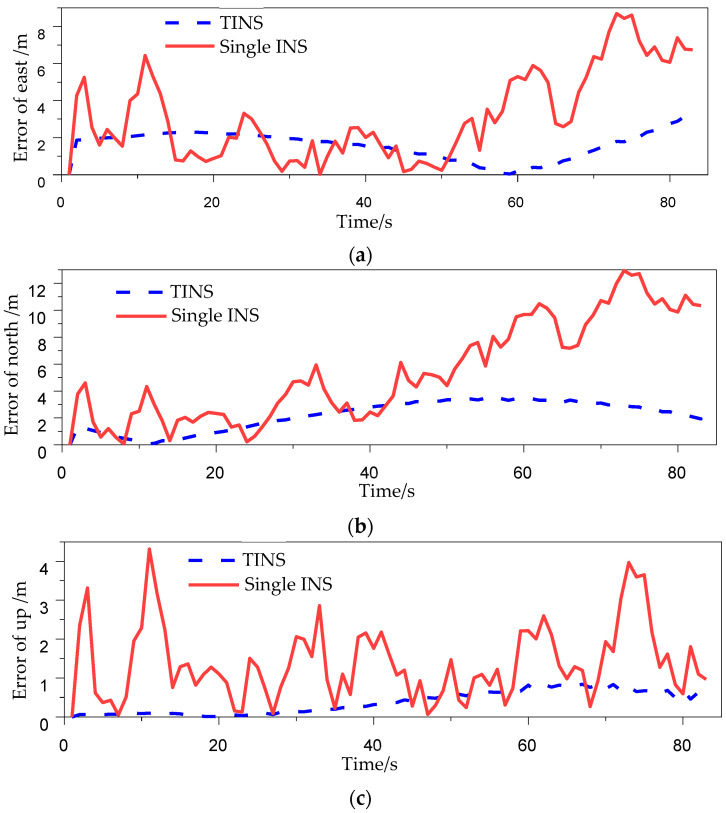
Error of shearer positioning. (**a**) Error of east. (**b**) Error of north. (**c**) Error of up.

**Figure 9 micromachines-14-01474-f009:**
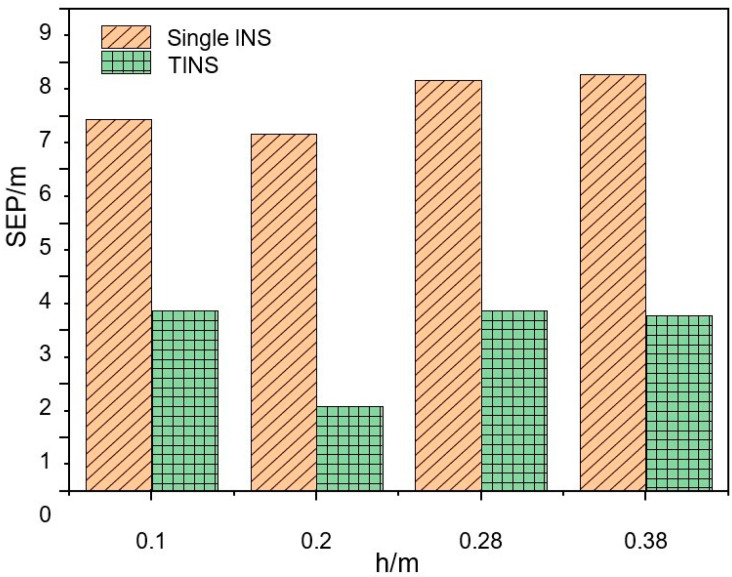
Comparison of positioning accuracy in the inter-INS distances influence experiments.

**Figure 10 micromachines-14-01474-f010:**
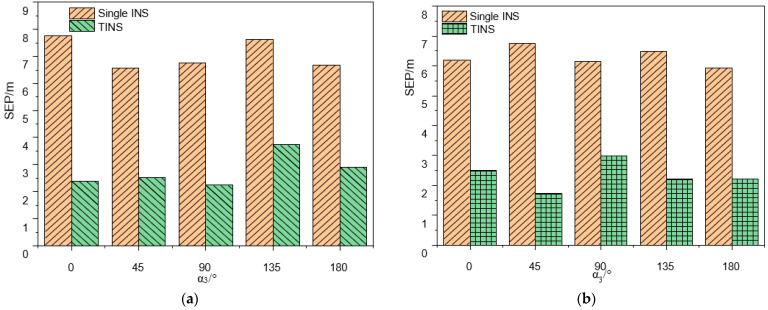
Comparison of positioning accuracy in the tri-INS plane spatial position influence experiments: (**a**) Positive plane; (**b**) Negative plane.

**Figure 11 micromachines-14-01474-f011:**
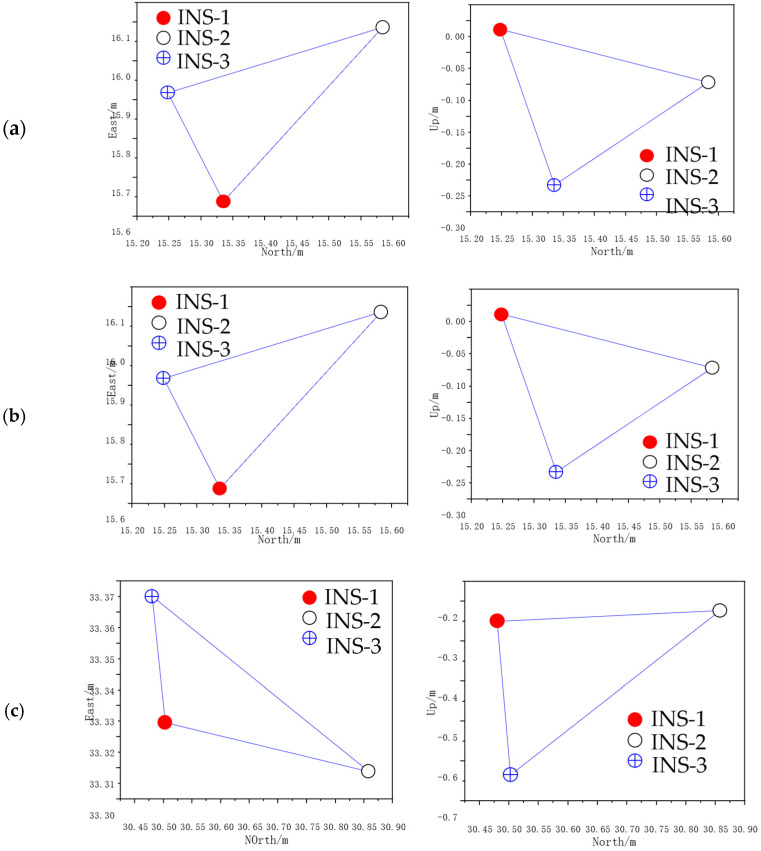
Projections of estimated triangle on east-north plane (left) and north-up plane (right): (**a**) 0 s; (**b**) 20 s; (**c**) 40 s.

**Figure 12 micromachines-14-01474-f012:**
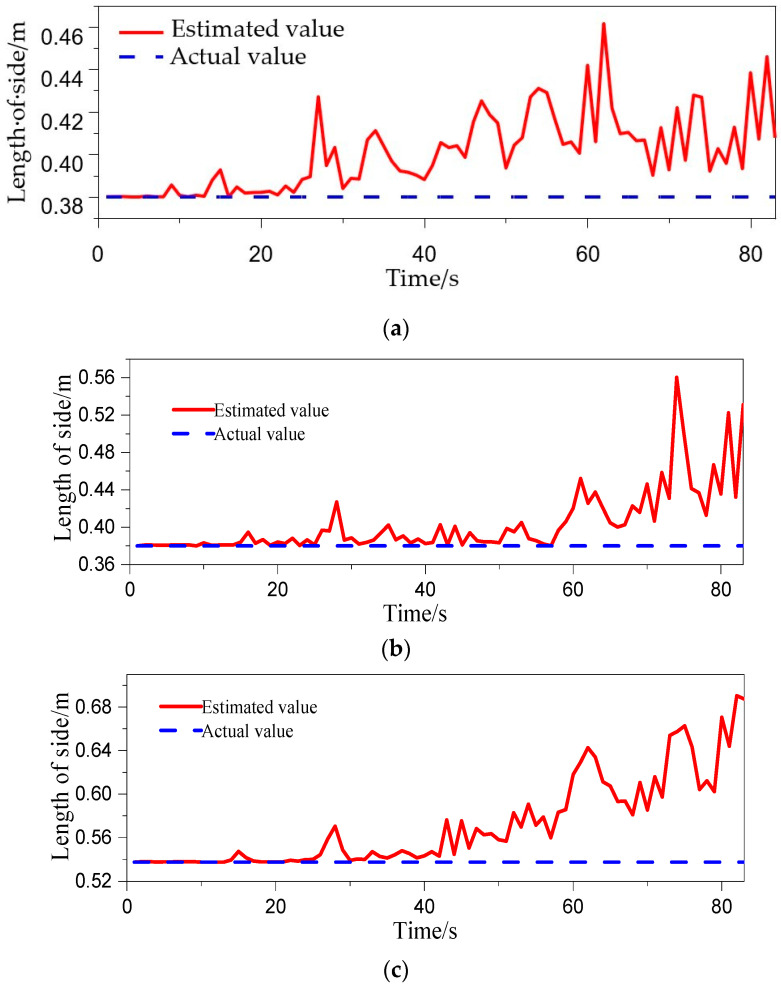
Variation of the side length of the estimated triangle. (**a**) r_12_; (**b**) r_13_; (**c**) r_23_.

**Table 1 micromachines-14-01474-t001:** Main performance indexes of gyroscopes of INS.

Measuring Range	Zero-Bias Stability	Distinguishability	In-Band Noise	Attitude Accuracy
±300°/s	≤18°/h	0.03°/s	0.3°/s	<0.3 deg (RMS)

**Table 2 micromachines-14-01474-t002:** Installation parameters table of experimental device.

	Inter-INS Distances Influence Experiments				Tri-INS Plane Spatial Position Influence Experiments									
					Positive Plane					Negative Plane				
No.	1	2	3	4	5	6	7	8	9	10	11	12	13	14
*r* _12_	0.1 m	0.2 m	0.28 m	0.38 m	0.38 m	0.38 m	0.38 m	0.38 m	0.38 m	0.38 m	0.38 m	0.38 m	0.38 m	0.38 m
*r* _13_	0.1 m	0.2 m	0.28 m	0.38 m	0.38 m	0.38 m	0.38 m	0.38 m	0.38 m	0.38 m	0.38 m	0.38 m	0.38 m	0.38 m
*α* _3_	90°	90°	90°	90°	0°	45°	90°	135°	180°	0°	45°	90°	135°	180°

## Data Availability

The data presented in this study are available on request from the corresponding author.

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
