# Peer review of "Experimental Research of Triple Inertial Navigation System Shearer Positioning"

_micromachines, 2023, doi:10.3390/mi14071474_

Round 1

Reviewer 1 Report

(1) The navigation system on shearer always works for a very long time (several hours). But the experiment time in fig.7 & fig.11 is too short, it cannot verify the effectiveness of the proposed method.

(2) The state selection in eq.(1) is too simple. Why not velocity, misalignment angle, gyro bias, acc bias listed as states?

(3) ‘The experimental device is composed of three INS with the same type and its manual turntable’, the main performance of the IMU should be presented.

(4) ‘The experimental device is composed of three INS with the same type and its manual turntable’, maybe this accuracy cannot meet the real shearer application.

(5) In fig.7, the results of TINS always improved over single-INS is suspicious. The authors may do more tests.

NA

Reviewer 2 Report

In view of the limited effect of the mainstream methods of improving the positioning accuracy of shearer, the Tri-INS positioning technology was adopted to improve the positioning accuracy of shearer in this paper. Furthermore, the influence of Inter-INS distances and positions of the three INS was analyzed in detail, the optimal installation parameters of the positioning device based on a variety of factors was proposed. However, there are still some problems in this paper that need to be improved further as follows: 
1. I wonder whether the authors could add some more references on factors of influencing INS precision in the introduction.
2. The author should emphasize the rational for using Multi-INS technology under mine in the introduction section.
3.  Probably the quality of the pictures in Figure 6 could be improved.
4. The first letter of ‘tri-INS’ and ‘multi-INS’ should be capitalized in this paper.
5. For better understanding the experiments process, the authors should provide flowchart for experiments.

Reviewer 3 Report

The submission is of high significance to intelligent and unmanned mining based on inertial navigation technology. However, 3 issues need to be addressed before acceptance for publication: 1) the composition of the 3 INS is required to be displayed together with the working mechanisms; 2) vibration in the real-world shall have severe impacts on the position/attitude  accuracy, which is yet to be discussed briefly; 3) the moving speed of the shearer on the positioning precision is yet to be discussed.

Extensive refinement of literal expresssion is required.

Round 2

Reviewer 1 Report

(1) The model in eq.(3) is too simple.

(2) I believe that it cannot improve the accuracy by using three of the same type of IMU in essential and it cannot apply for real system.

NA

Reviewer 3 Report

The re-submission has been somewhat improved.  After refinement of the literal expression and figures, the manuscript may be accepted for publication.

The quality of the English language is acceptable for current version.

Author Response

Dear Editors and Reviewers:

The main corrections in the paper and the responds to the reviewer's comments are as follows:

The reviewer’s comment: The re-submission has been somewhat improved.  After refinement of the literal expression and figures, the manuscript may be accepted for publication.

Response:In the revised version, We have done our best to refine the wording and some of the figures. These changes will not influence the content and framework of the paper.

Sincere thanks to editors and reviewers !